# Reviving product states in the disordered Heisenberg chain

Henrik Wilming [1] ✉, Tobias J. Osborne [1], Kevin S. C. Decker [2] & Christoph Karrasch[2]

When a generic quantum system is prepared in a simple initial condition, it typically equilibrates toward a state that can be described by a thermal ensemble. A known exception is localized systems that are non-ergodic and do not thermalize; however, local observables are still believed to become stationary. Here we demonstrate that this general picture is incomplete by constructing product states that feature periodic high-fidelity revivals of the full wavefunction and local observables that oscillate indefinitely. The system neither equilibrates nor thermalizes. This is analogous to the phenomenon of weak ergodicity breaking due to many-body scars and challenges aspects of the current phenomenology of many-body localization, such as the logarithmic growth of the entanglement entropy. To support our claim, we combine analytic arguments with large-scale tensor network numerics for the disordered Heisenberg chain. Our results hold for arbitrarily long times in chains of 160 sites up to machine precision.

When a large, closed, interacting quantum many-body system is initialized in a simple initial condition, it typically approaches a state that is stationary when only observed with coarse-grained (e.g., local) observables−the system *equilibrates*[1,2]. In addition, the stationary state of the coarse-grained observables is often well-described by statistical (e.g., canonical) ensembles−the system *thermalizes*[2–5]. While thermalization is a generic phenomenon and aids the theoretical description, it is not inevitable. One of the most intensely debated exceptions is that of *many-body localization* (MBL), which is realized in interacting quantum models with a sufficiently strong disorder potential[6–9]. Systems exhibiting MBL provide generic examples of non-ergodic systems that fail to thermalize due to a memory of the local initial conditions, yet they are still equilibrating[10,11]. Other key features of MBL phases include an unbounded growth of the entanglement during quantum quenches[12–14] and peculiar transport properties[15–17]. There are now a variety of experimental realizations exhibiting signatures of MBL, including cold atoms[18,19] and photonic systems[20].

The existence of MBL as a stable phase of matter has recently been questioned, and it has been suggested that thermalization actually eventually occurs[21–25]. However, it is fair to say that a conclusive picture has not yet emerged[26–31]. A key obstacle is that many studies are based on an exact diagonalization of small systems and might thus not be representative of the behavior in the thermodynamic limit[32,33]. Approaching the problem from the perspective of quantum avalanches has been a major recent direction[34–42].

Another exception to the rule of equilibration and thermalization was recently discovered: In so-called *many-body scarred* systems, there exists a relatively small set of initial product states that may show indefinite revivals of the full many-body wavefunction. When the system is initialized in such an initial state, all physical observables (including local ones) show periodic oscillations, and the system neither thermalizes nor equilibrates[43–50]. The revivals of the wavefunction are connected to the existence of a small set of high-energy eigenstates that exhibit atypically low entanglement, dubbed "quantum (many-body) scars". Conversely, if all energy eigenstates are sufficiently entangled, then initial product states generically equilibrate[51–53].

In fact, MBL systems also exhibit quantum many-body scarring, and they do so in a most dramatic way: Not just a few, but all high-energy eigenstates have atypically low entanglement since

[1]Leibniz Universität Hannover, Appelstraße 2, 30167 Hannover, Germany. [2]Technische Universität Braunschweig, Institut für Mathematische Physik, Mendelssohnstraße 3, 38106 Braunschweig, Germany. ✉e-mail: henrik.wilming@itp.uni-hannover.de

the entanglement entropy features an area law[54–59] instead of a volume law. (This is the generic situation in interacting systems[60–68]).

To summarize, many-body scarred systems host a few slightly entangled eigenstates, and these can be sufficient for a complete breakdown of equilibration in certain initial product states. Conversely, all energy eigenstates in MBL systems are low-entangled. This leads to a natural question: Can MBL systems also host initial product states that show high-fidelity revivals of the wavefunction with corresponding local observables that oscillate indefinitely?

If the answer to this question is "yes", then—contrary to current belief—MBL systems do not generally equilibrate from product states and hence also do not thermalize. Moreover, a further hallmark feature of MBL, namely the slow (logarithmic) but unbounded growth of the entanglement entropy, would be violated for these particular initial conditions.

It is, however, unclear how to approach this problem and how to find such initial conditions for a given MBL Hamiltonian. In particular, there are two key difficulties to be overcome: (1) The product state might have to be fine-tuned to the details of the Hamiltonian, such as the disorder configuration. However, the set of product states is a continuum, so we cannot simply search through all of them. Furthermore, we cannot exploit algebraic structures (such as symmetries) to guide us; (2) Even given a candidate's initial state, how could we make sure that it does not equilibrate? In principle, the revival could happen at arbitrary long times, which cannot be accessed analytically or numerically (even for MBL systems).

In this work, we overcome these difficulties and demonstrate that one can find initial product states featuring high-fidelity revivals and local observables that oscillate indefinitely. We combine analytical arguments with state-of-the-art tensor network calculations. Importantly, our approach works for arbitrarily long times, and we can treat systems of up to 160 sites with machine precision.

## Results

We focus on the paradigmatic disordered spin-1/2 Heisenberg model on $L$ lattice sites,

$$\hat{H} = -\sum_{j=1}^{L-1} S_j \cdot S_{j+1} + \sum_{j=1}^{L} h_j \hat{S}_j^{(z)}, \qquad (1)$$

where $S_j = (\hat{S}_j^{(x)}, \hat{S}_j^{(y)}, \hat{S}_j^{(z)})^{\top}$ is the vector of spin-1/2 angular momentum operators at site $j$. The local magnetic fields $h_j \in [-W, W]$ are sampled independently from a uniform distribution; $W$ is the disorder strength. Exact diagonalization of small systems predicts a crossover from an ergodic to an MBL phase around $W \sim 3.5$[33]. In the main part of this work, we set $W = 8$.

First, we show that if we can find two eigenstates whose superposition is well approximated by a product state, then one can construct a local observable that oscillates indefinitely with an amplitude that is lower-bounded by a *certified amplitude* $A_{\text{cert.}}$ ("Results: Locally oscillating product states").

Second, we use large-scale tensor network numerics to construct such eigenstates for the disordered Heisenberg chain ("Results: Numerical construction"). We present data for systems of up to $L = 160$ sites and, up to machine precision, provide a rigorous certificate for the indefinite oscillations of a local observable ("Results: Main results").

Lastly, we present theoretical arguments suggesting that large systems may, in fact, host a finite density of locally oscillating excitations ("Results: Multiple localized dynamical oscillations").

Our results are illustrated in Fig. 1. To keep the discussion concise, we delegate most technical details to the "Methods" section and the Supplementary Information.

### Locally oscillating product states

Let us consider two eigenstates $|E_1\rangle$ and $|E_2\rangle$. Their time-evolved equal superposition

$$|\Psi(t)_{\pm}\rangle = \frac{1}{\sqrt{2}}\left(e^{-iE_1 t}|E_1\rangle \pm e^{-iE_2 t}|E_2\rangle\right) \qquad (2)$$

shows perfect revivals at even multiples of the period $\tau = \pi/(E_1 - E_2)$. Now suppose there is a product state

$$|\Phi(0)_{\pm}\rangle = |\phi_{\pm}^{(1)}\rangle \otimes \cdots \otimes |\phi_{\pm}^{(L)}\rangle \qquad (3)$$

that approximates $|\Psi_{\pm}(0)\rangle$ in the sense that its overlap fulfills $F_{\pm}^2 = |\langle\Psi(0)_{\pm}|\Phi(0)_{\pm}\rangle|^2 \geq 1 - \epsilon$ with $\epsilon$ small. This implicitly defines the local quantum states $|\phi_{\pm}^{(k)}\rangle$. The simple but key observation of our approach is that the time-evolved state $|\Phi(t)_{\pm}\rangle = \exp(-i\hat{H}t)|\Phi(0)_{\pm}\rangle$ will necessarily also show high-fidelity revivals:

$$|\langle\Phi(0)_{\pm}|\Phi(2k\tau)_{\pm}\rangle|^2 \geq 1 - 4\epsilon \qquad (4)$$

for any integer $k$. Moreover, let $j = \text{argmin}_k|\langle\phi_+^{(k)}|\phi_-^{(k)}\rangle|$. Then the observable

$$\hat{A} = \mathbb{1} \otimes \left(|\phi_+^{(j)}\rangle\langle\phi_+^{(j)}| - |\phi_-^{(j)}\rangle\langle\phi_-^{(j)}|\right) \otimes \mathbb{1} \qquad (5)$$

is supported on a single site, and its time-dependent expectation value in the state $|\Phi_+(t)\rangle$ oscillates with period $\tau$:

$$\left|\langle\Phi_+|\hat{A}(2k\tau)|\Phi_+\rangle - \langle\Phi_+|\hat{A}((2k+1)\tau)|\Phi_+\rangle\right| \geq A_{\text{cert.}} \qquad (6)$$

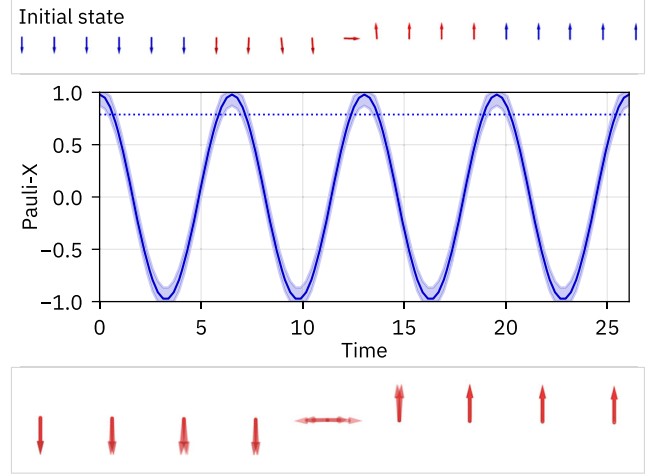

**Fig. 1 | Indefinitely oscillating spins.** Top: The disordered Heisenberg chain ($L = 20$) is initialized in a deformed domain wall product state that has an overlap >0.994 with a superposition of two energy eigenstates. Middle: Under the unitary time evolution, the local spins remain almost uncorrelated and start to oscillate in the region around the domain wall interface. The solid line shows the expectation value of the Pauli-X observable $\langle 2\hat{S}_j^x \rangle$ at the center spin in the superposed energy eigenstates. The dynamics of the actual product state is within the associated shaded regions due to its large overlap with the superposition of eigenstates. The blue dotted line indicates the certified amplitude, which provides a lower bound for the magnitude of the oscillations in the infinite-time limit. Bottom: Overlay of different snapshots in time of the expectation values of the local spin operators around the domain wall interface, visualized as arrows within their respective Bloch spheres.

for any integer $k$. $\hat{A}(t)$ refers to the Heisenberg picture. The certified amplitude $A_{\text{cert.}}$ is given by

$$A_{\text{cert.}} = \max\{1 - f^2 - 2\sqrt{(1-f^2)\epsilon}, 0\}, \qquad (7)$$

where $f^2 = \min_j |\langle \phi_+^{(j)} | \phi_-^{(j)} \rangle|^2$ measures the minimal local overlap between $|\Phi(0)_+\rangle$ and $|\Phi(0)_-\rangle$ (assuming that each $|\phi_\pm^{(j)}\rangle$ is normalized). A detailed proof can be found in "Methods: Certified amplitudes".

## Numerical construction

As a next step, we demonstrate how to find pairs of energy eigenstates whose equal superpositions are well approximated by product states. It is reasonable to hypothesize that such states must have a low entanglement with respect to any bipartition. Therefore we performed a structured search on small systems using exact diagonalization and targeting energy eigenstates whose sublattice entanglement entropy ($ABABAB\ldots$-bipartition) is small; see Supplementary Material for more details. Targeting small sublattice entanglement is a heuristic choice motivated by the following considerations: (1) Product states have vanishing sublattice entanglement entropy and therefore any state sufficiently close to a product state should have small sublattice entanglement and (2) even generic translationally invariant matrix-product states (MPS)[69,70], which are commonly considered to be low-entangled, have extensive sublattice entanglement entropies[71]. Therefore small sublattice entanglement heuristically indicates an amount of entanglement that is small even compared to MPS. Our preliminary analysis showed that pairs of energy eigenstates whose equal superpositions are well approximated by product states exist and that one class of them comes in the form of deformed domain walls (see Fig. 1). This knowledge then allows us to devise an efficient tensor-network based algorithm to study large systems, which we now briefly explain (further details may be found in "Methods: Details of our numerical method").

At sufficiently strong disorder, the eigenstates of $\hat{H}$ feature an area-law entanglement and may be represented faithfully as MPS[57], whose explicit representation can be determined using the DMRG-X algorithm[72]. The algorithm starts with a "seed" state $|m_1\rangle \otimes \cdots \otimes |m_L\rangle$, where $|m_j\rangle \in \{|\uparrow\rangle, |\downarrow\rangle\}$ denote the eigenstates of $\hat{S}_j^{(z)}$. These seeds are the eigenstates of $\hat{H}$ in the limit of $W \to \infty$. DMRG-X then iteratively determines an (approximate) eigenstate at finite $W$ that is, in a sense, closest to the initial seed. The main numerical control parameter is the so-called bond dimension $\chi$, which we choose so that high-energy eigenstates are obtained up to machine precision.

In our case, we find the energy eigenstates $|E : k\rangle$ associated with seeds in domain-wall form

$$|\mathrm{dw} : k\rangle = \underbrace{|\downarrow\rangle \otimes \cdots \otimes |\downarrow\rangle}_{k\,\text{times}} \otimes |\uparrow\rangle \otimes \cdots \otimes |\uparrow\rangle. \qquad (8)$$

We then form the superposition of the energy eigenstates resulting from neighboring domain walls,

$$\left|\Psi_\pm^{(k)}\right\rangle = \frac{1}{\sqrt{2}}\left(|E : k\rangle \pm |E : k+1\rangle\right), \qquad (9)$$

and finally construct their product-state approximation $|\Phi_\pm^{(k)}\rangle$. This allows us to calculate the certified amplitude $A_{\text{cert.}}$ of Eq. (7). All of these operations can be implemented efficiently and accurately in the MPS representation (see "Methods: Details of our numerical method" for further details). We stress that at this point, it is not clear why the states $|\Psi_\pm^{(k)}\rangle$ should be close to product states apart from the fact that we found revolving product states with a similar structure in our small-scale exact-diagonalization numerics (see Supplementary Material). Our main results in the next section show that for domain-wall seeds, closeness to a product state is indeed a generic case for sufficiently strong disorder. This, in turn, immediately implies the non-equilibrating behavior for the associated product states.

## Main results

In Fig. 2, our aggregated numerical data for the certified amplitude at varying system sizes up to $L = 160$ and at a disorder strength $W = 8$ with 100 disorder realizations per system size is depicted (the corresponding fidelities are discussed in Supplementary Note 1). We find median certified amplitudes of the order of 0.7, essentially independent of the system size, with decreasing fluctuations as $L$ increases. Moreover, the maximum certified amplitudes for domain-wall states with interface in the middle half of the system (sites $k = L/4$ to $k = 3L/4$) slowly increase with system size, with all sampled realizations reaching $A_{\text{cert.}} > 0.91$ for $L = 160$. The restriction to states with the interface in the middle half of the system excludes states that can be interpreted as being close to single-particle excitations (see below and Supplementary Note 2). We emphasize that the certified amplitude provides a lower bound to the magnitude of the oscillations of $\hat{A}$ and that there may exist local operators which oscillate with even higher amplitude.

In a nutshell, Fig. 2 conclusively demonstrates the (generic) existence of initial product states that host high-fidelity revivals

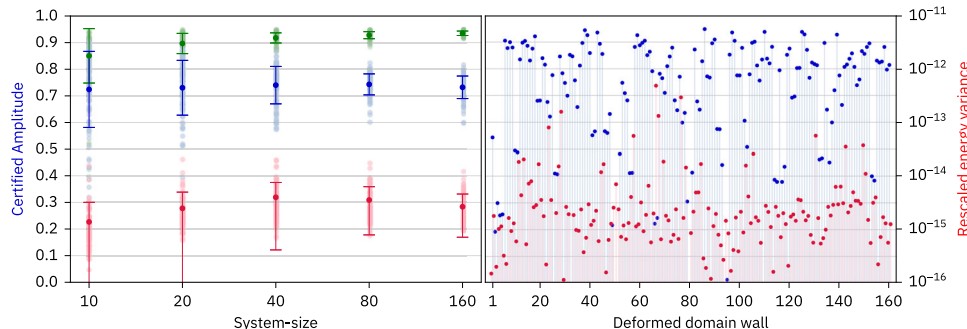

**Fig. 2 | Certified amplitudes.** Left: Median (light blue) and maximum (light green) of the certified amplitudes that provide a lower bound for the infinite-time oscillations of a local spin observable in a product state corresponding to a deformed domain (the median and maximum are taken w.r.t. the different positions of the domain wall; the maximum is restricted to domain walls with an interface in the middle half of the system, i.e., sites $L/4$ to $3L/4$). We present aggregated data for 100 disorder realizations per system size with disorder strength $W = 8$ (each point corresponds to one disorder realization). Dark points with error bars show the mean and standard deviation of the associated values. We also plot the median rescaled energy variances $\sigma^2/E^2$ of the eigenstates determined using the DMRG-X algorithm (light red dots) together with their mean and associated variance (dark red). Right: Certified amplitudes (blue) as well as the rescaled energy variances of the two associated eigenstates (red) for all deformed domain walls and a single disorder realization.

and the existence of local, indefinitely oscillating observables in a system of up to 160 sites. The overall shape of these product states is in the form of two domain walls separated by a spin pointing roughly in $\pm x$-direction at their interface. Moving away from the interface, the spins still point away from their original $\pm z$-directions but with decreasing components in the $x-y$ plane. This is visualized in Fig. 1. As a side remark, we mention that the Hamiltonian $\hat{H}$ may also be interpreted as a Hamiltonian of interacting fermions by a Jordan-Wigner transformation. However, in this picture, the parity super-selection rule forbids our reviving product states since they correspond to superpositions of states with different fermion-number parity.

The fact that we find oscillating deformed domain walls is particularly interesting since previous results indicate that the bare domain-wall states $|dw : k\rangle$ approach a steady state with a smeared-out interface, a process known as domain-wall melting[73]. An interface spin pointing away from the $z$-axis therefore protects against this mechanism.

Since the DMRG-X algorithm outputs the energy eigenstates $|E : k\rangle$ as MPS, we can compute the expectation values $\langle \Psi_{\pm}^{(k)}|\hat{B}(t)|\Psi_{\pm}^{(k)}\rangle$ of any local operator $\hat{B}$ exactly for arbitrary times $t$ (see "Methods: Details of our numerical method"). This, in turn, allows us to quantitatively estimate the finite-time expectation value $\langle \Phi_{\pm}^{(k)}|\hat{B}(t)|\Phi_{\pm}^{(k)}\rangle$ for any local $\hat{B}$, which is useful since the certified amplitude only provides a lower bound for the oscillations of the specific local observable $\hat{A}$ (yet at infinite times). In Fig. 1, we visualize this for $\hat{S}^{(x)}$, which is not strictly identical with the observable $\hat{A}$.

Besides the deformed domain-wall states, there exists a second set of reviving product states that exhibit local oscillations. However, these can be interpreted as a single-particle phenomenon arising from Anderson localization and exist irrespective of the strength of the term $\hat{S}_j^{(z)}\hat{S}_{j+1}^{(z)}$, see Supplementary Note 2.

In Supplementary Note 3, we further provide numerical data for the certified amplitude and various disorder strength in the range $W = 0.5$ to $W = 8$. One can identify a crossover from an ergodic system to a localized system.

### Multiple localized dynamical oscillations

Our numerical data clearly demonstrates that product states with high-fidelity revivals and locally oscillating observables exist for the disordered Heisenberg model at sufficiently strong disorder. However, our approach only yields states with single dynamical excitations. We now explain our construction qualitatively from a different point of view and argue for the existence of product states with a finite density of such dynamical excitations.

Since the product states $|m_1\rangle \otimes \cdots \otimes |m_L\rangle$ and the energy eigenstates $|E_j\rangle$ both provide an orthonormal basis of the Hilbert space, there exists a unitary mapping $\hat{U}$ between the two. The mapping is

believed to be quasi-local[9,55,59,74], which implies that it maps local operators to operators whose support is still localized in space with potentially (sub-)exponential tails. As a simplified model for this situation, we may think of $\hat{U}$ as a local quantum circuit of finite depth and composed of gates that only couple nearest neighbors. At the same time, the Hamiltonian $\hat{H}$, and therefore also the unitary $\hat{U}$, has the states $|\downarrow\rangle \otimes \cdots \otimes |\downarrow\rangle$ and $|\uparrow\rangle \otimes \cdots \otimes |\uparrow\rangle$ as eigenstates. In the bulk of a large region of spins, all pointing upward or downward, $\hat{U}$ must therefore act like the identity. Quasi-locality immediately implies that $|E : k\rangle = \hat{U}|dw : k\rangle$ only contains a localized, static excitation around the domain-wall interface, see Fig. 3. The superposition $|\Psi_{\pm}^{(k)}\rangle$, which shows perfect revivals, must hence support an operator localized around $k$ whose expectation value oscillates in time, i.e., a dynamical, localized excitation.

This discussion suggests that in a large system, we may construct multiple domain walls separated by dynamical, localized excitations as long as the size of each domain wall is sufficiently large. A finite density of local dynamical excitations should hence, in principle, be possible. However, each such excitation doubles the number of energy eigenstates that need to be superposed, and the cost of simulating such situations scales exponentially with the number of excitations. In Supplementary Fig. 3, we provide proof-of-principle numerics in a system of size $L = 80$ with up to three excitations, supporting the general argument described above; see Supplementary Note 4 for more details.

## Discussion

Anderson's discovery that a random potential can have strong effects on the transport properties of a free quantum particle was a milestone in condensed matter physics. In the last decade, the fate of Anderson localization in the presence of two-body interactions has received significant attention, and it is believed that generic non-ergodic—so-called many-body localized—systems exist. A key feature of these systems is that simple initial states do not thermalize while local observables still equilibrate. (Some comments on the recent controversy about the existence of the MBL phase can be found in the introduction).

In this work, we provided analytical and numerical arguments that this picture is not correct and that one can construct simple product states that show a complete absence of both thermalization and equilibration. The full many-body wavefunction exhibits high-fidelity revivals, and local spin operators oscillate with large amplitudes. We demonstrated this for the prototypical disordered Heisenberg chain via large-scale tensor network numerics for systems of up to $L = 160$ sites. Our results hold for arbitrary long times up to machine precision.

We also argued that multiple such localized dynamical excitations exist in large systems, giving rise to a picture reminiscent of "Hilbert-space fragmentation" in systems with quantum many-

Single dynamical excitation

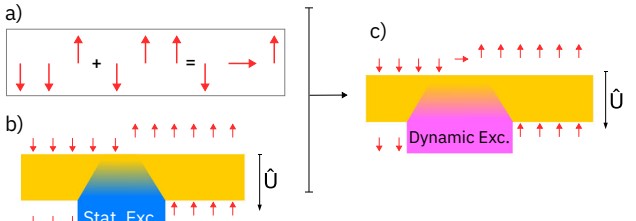

Multiple dynamical excitations

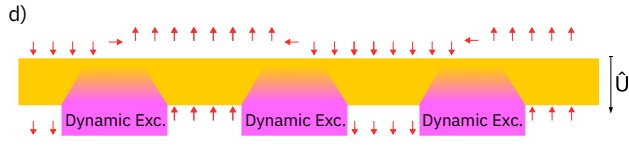

**Fig. 3 | Qualitative picture for dynamical excitations. a** Two neighboring domain walls superpose to a domain wall separated by a spin pointing in $\pm x$-direction. **b** At sufficiently strong disorder, the unitary transformation $\hat{U}$ that maps eigenstates of $\hat{S}_j^{(z)}$ to energy eigenstates is (quasi-)local and leaves the states with all spins pointing up or down invariant. Acting on a domain wall, it therefore yields an energy

eigenstate with localized static excitation. **c** Acting on the two superposed, neighboring domain walls, the unitary $\hat{U}$ yields a localized dynamical excitation with perfect revivals. **d** If several sufficiently large domain walls are separated by spins pointing in $\pm x$-directions, acting with $\hat{U}$ yields a finite density of localized, dynamical excitations.

body scars arising from kinematic constraints (see ref. [75] and references therein). Similar results have been found for systems showing so-called "Stark many-body localization", which are translationally invariant systems reproducing much of the MBL phenomenology[76–79]. In this case, local oscillating observables can be proven to exist[80] using the concept of dynamical symmetries[81]. In contrast to these disorder-free systems, in our case, all of these features depend on the precise disorder realization. Therefore we do not expect a clean, emergent algebraic structure associated with the subspace spanned by states with multiple excitations, but we also cannot rule out such a structure. We therefore leave a detailed investigation for future work.

Basic MBL phenomenology has been successfully demonstrated experimentally using ultra-cold Fermions in optical lattices[18] and trapped ions[19]. Due to the efficient nature of our algorithm, it is, in principle, possible to calculate the non-equilibrating product states on the fly given a (quasi-)random disorder realization, even for relatively large system sizes. Since the preparation of deformed domain walls only requires precise single-site addressing for a few of the spins (with the remaining spins being in large blocks of all up and all down), it should therefore be possible to observe the resulting revivals in present-day or near-future experiments.

Our results were made possible by developing a method to systematically find fine-tuned initial product states. So far, no general and efficient method exists to find product states that resist equilibration and thermalization in general interacting many-body systems. Devising such an approach to study models that are currently believed to be thermalizing is a fruitful future direction.

## Methods
### Certified amplitudes
We derive Eq. (4) and show how to determine the local spin observable $\hat{A}$ that oscillates with the certified amplitude given in Eq. (7). We make use of the general relation

$$|\langle \Psi | \Phi \rangle|^2 = 1 - D[\hat{\Psi}, \hat{\Phi}]^2 \tag{10}$$

between the fidelity and the trace distance $D$ for two pure states, where we use the notation $\hat{\Psi} = |\Psi\rangle \langle \Psi|$. The trace distance fulfills the triangle inequality:

$$|\langle \Phi_+(0) | \Phi_+(t_{2k}) \rangle|^2$$
$$\geq 1 - (D[\hat{\Phi}_+(0), \hat{\Psi}_+(0)] + D[\hat{\Phi}_+(t_{2k}), \hat{\Psi}_+(0)])^2, \tag{11}$$

where $t_n = n\tau$. Employing $\hat{\Psi}_+(t_{2k}) = \hat{\Psi}_+(0)$ as well as the fact that the trace distance is invariant under unitary transformations and hence under time-translation, we get $D[\hat{\Phi}_+(0), \hat{\Psi}_+(0)] = D[\hat{\Phi}_+(t_{2k}), \hat{\Psi}_+(0)] = \sqrt{1 - F_+^2}$. This yields

$$|\langle \Phi_+(0) | \Phi_+(t_{2k}) \rangle|^2 \geq 1 - 4(1 - F_+^2) \geq 1 - 4\epsilon, \tag{12}$$

where we used the assumption $F_+^2 \geq 1 - \epsilon$.

We now turn to the operator $\hat{A}$ and its certified amplitude. Let $j = \text{argmin}_k |\langle \phi_+^{(k)} | \phi_-^{(k)} \rangle|$ be the site where the local overlap between $|\Phi_-\rangle$ and $|\Phi_+\rangle$ is minimized so that $f = |\langle \phi_+^{(j)} | \phi_-^{(j)} \rangle|$. We then define $\hat{A}$ as

$$\hat{A} = \mathbb{1} \otimes \cdots \otimes \left( |\phi_+^{(j)}\rangle \langle \phi_+^{(j)}| - |\phi_-^{(j)}\rangle \langle \phi_-^{(j)}| \right) \otimes \cdots \otimes \mathbb{1} \tag{13}$$

The operator-norm of $\hat{A}$ is given by $\| \hat{A} \| = \sqrt{1 - f^2}$. For any observable $\hat{X}$ and any two density matrices $\hat{\rho}$ and $\hat{\sigma}$ it holds that

$$|\text{Tr}[\hat{X}\hat{\rho}] - \text{Tr}[\hat{X}\hat{\sigma}]| \leq \| \hat{X} \| \, D[\hat{\rho}, \hat{\sigma}]. \tag{14}$$

Using $\hat{\Psi}_-(0) = \hat{\Psi}_+(t_{2k+1})$, we therefore find

$$|\text{Tr}[\hat{A}\hat{\Phi}_+(t_{2k+1})] - \text{Tr}[\hat{A}\hat{\Psi}_-(0)]|$$
$$\leq \| \hat{A} \| \, D[\hat{\Phi}_+(t_{2k+1}), \hat{\Psi}_+(t_{2k+1})] \tag{15}$$

$$= \sqrt{(1 - f^2)(1 - F_+^2)} \tag{16}$$

$$\leq \sqrt{(1 - f^2)\epsilon}, \tag{17}$$

where we used $F_\pm^2 \geq 1 - \epsilon$. Similarly,

$$|\text{Tr}[\hat{A}\hat{\Phi}_-(0)] - \text{Tr}[\hat{A}\hat{\Psi}_-(0)]| \leq \sqrt{(1 - f^2)\epsilon}. \tag{18}$$

The triangle inequality then yields

$$|\text{Tr}[\hat{A}\hat{\Phi}_+(t_{2k+1})] - \text{Tr}[\hat{A}\hat{\Phi}_-(0)]| \leq 2\sqrt{(1 - f^2)\epsilon}, \tag{19}$$

and a similar calculation shows

$$|\text{Tr}[\hat{A}\hat{\Phi}_+(t_{2k})] - \text{Tr}[\hat{A}\hat{\Phi}_+(0)]| \leq 2\sqrt{(1 - f^2)\epsilon}. \tag{20}$$

Since $f = |\langle \phi_+^{(j)} | \phi_-^{(j)} \rangle|$, we further have

$$\text{Tr}[\hat{A}\hat{\Phi}_+(0)] = 1 - f^2, \quad \text{Tr}[\hat{A}\hat{\Phi}_-(0)] = f^2 - 1. \tag{21}$$

In total, we find

$$\text{Tr}[\hat{A}\hat{\Phi}_+(t_{2k})] - \text{Tr}\left[\hat{A}\hat{\Phi}_+(t_{2m+1})\right]$$
$$= \text{Tr}\left[\hat{A}\hat{\Phi}_+(0)\right] + \text{Tr}\left[\hat{A}(\hat{\Phi}_+(t_{2k}) - \hat{\Phi}_+(0))\right]$$
$$- \text{Tr}\left[\hat{A}\hat{\Phi}_-(0)\right] + \text{Tr}\left[\hat{A}(\hat{\Phi}_-(0) - \hat{\Phi}_+(t_{2m+1}))\right] \tag{22}$$
$$\geq (1 - f^2) - 2\sqrt{(1 - f^2)\epsilon} - (f^2 - 1) - 2\sqrt{(1 - f^2)\epsilon}$$
$$= 2(1 - f^2 - 2\sqrt{(1 - f^2)\epsilon})$$

for any $k, m \in \mathbb{N}$.

### Details of our numerical method
We use a custom implementation of the DMRG-X algorithm, which takes into account the $U(1)$-symmetry of $\hat{H}$ (the Hamiltonian commutes with the total magnetization in $z$-direction). The ensuing time evolution of a state which is not an eigenstate of the total magnetization is computed exactly; see below.

The DMRG-X algorithm provides one way to find MPS representations of excited eigenstates in disordered systems. It starts with an initial MPS called the "seed" (which in our case is a product state on the basis of $\hat{S}_j^{(z)}$) and iteratively updates each tensor of the MPS by sweeping through the chain. This is analogous to a ground state calculation, but instead of minimizing the energy in each update step, one picks the eigenstate of the local Hamiltonian that maximizes the overlap with the previous MPS. The bond dimension is increased every 20 sweeps (see Fig. 4); we use values $\chi = 2, 4, 8, 16, 24, 32$ for our main data. The algorithm terminates once the rescaled energy variance $\sigma^2/E^2$ has fallen to at least $10^{-12}$ ($E$ and $\sigma$ are the bare energy and standard deviation of energy, respectively). As indicated in Fig. 2, we often even find rescaled energy variances below $10^{-14}$. In Table 1, we show how often it is not possible to reach convergence with a maximum bond dimension of $\chi = 32$ in all the calculations resulting in our main result Fig. 2. One should note that for a system of size $L = 10$, any state can be encoded with a bond dimension $\chi = 32$; however the absolute energy variance $\sigma^2$ can reach machine precision, while the rescaled $\sigma^2/E^2$ can

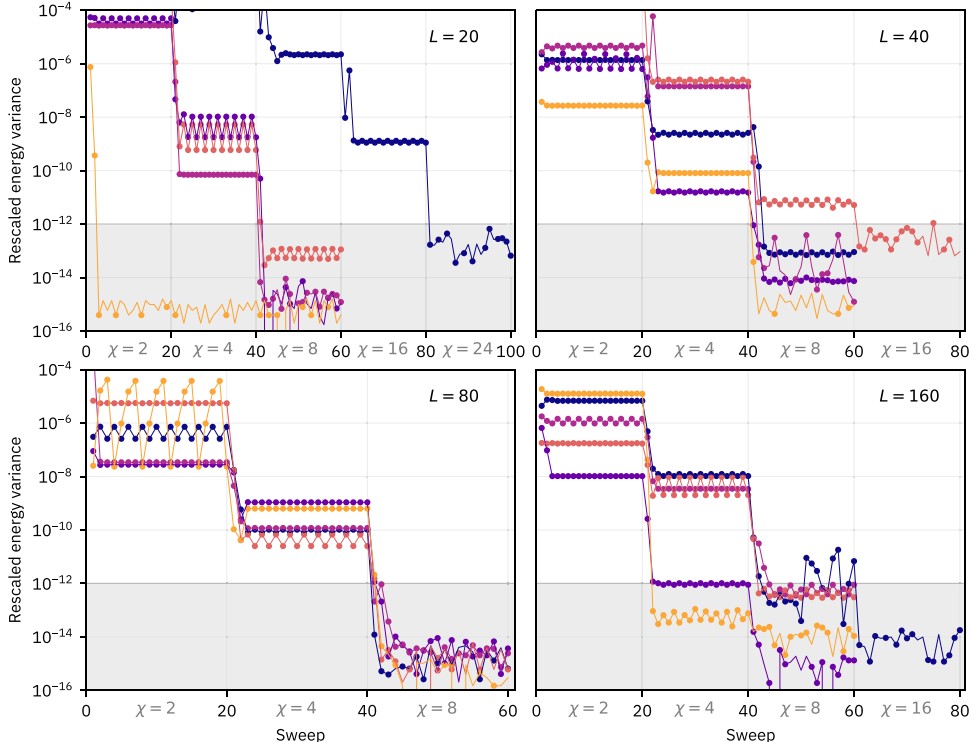

**Fig. 4 | Convergence of the DMRG-X algorithm.** Rescaled energy variance $\sigma^2/E^2$ along the DMRG-X sweeps for five DMRG-X runs (different initial seeds and disorder realizations) randomly chosen from the full dataset used for Fig. 2. The different panels correspond to system-sizes $L = 20, 40, 80, 160$ as indicated. The lines show the absolute value $|\sigma^2/E^2|$; missing dots correspond to negative signs (see the main text for details). After each 20 sweeps, the bond dimension $\chi$ is increased.

still be larger than $10^{-12}$ if $E$ is smaller than unity. Table 1 contains 6 such states ($L = 10$, $\chi = 32$).

Due to numerical rounding errors, the energy variance $\sigma^2$ may be negative when the calculation has converged to machine precision, even though variances are always positive semi-definite. In such cases, one observes final fluctuations with the same magnitude but differing signs, clearly signaling that the result should be interpreted as zero; see Fig. 4 for examples.

As shown in Supplementary Note 5, a pure state with energy variance $\sigma^2$ behaves as an eigenstate for time scales at least of the order of $1/\sigma$. Hence, the small threshold for the rescaled energy variance of $10^{-12}$ that we use guarantees on its own that all our conclusions remain valid for a time, at least of the order of $10^6$ (in the chosen units). In Fig. 5, we nevertheless also provide a comparison of the DMRG-X deformed domain-wall states with the closest eigenstates obtained from exact diagonalization for system sizes up to $L = 14$, showing excellent agreement in terms of fidelity.

**Finding the product-state approximation.** We now explain how to find a product-state approximation to a superposition $|\Psi_\pm(0)\rangle$. Denote by $\hat{\rho}_\pm^{(j)}$ the reduced density matrix at site $j$ in the state $|\Psi_\pm(0)\rangle$. As with any spin-1/2 density matrix, it may be written as

$$\hat{\rho}_\pm^{(j)} = \frac{1}{2}\mathbb{1} + r_\pm^{(j)} \cdot S_j, \qquad (23)$$

where $r_\pm^{(j)}$ is the vector that collects the expectation values of the local Pauli operators,

$$r_\pm^{(j)} = 2\begin{pmatrix} \langle\Psi_\pm(0)|\hat{S}_j^{(x)}|\Psi_\pm(0)\rangle \\ \langle\Psi_\pm(0)|\hat{S}_j^{(y)}|\Psi_\pm(0)\rangle \\ \langle\Psi_\pm(0)|\hat{S}_j^{(z)}|\Psi_\pm(0)\rangle \end{pmatrix}. \qquad (24)$$

The reduced density matrix is pure if and only if $r_\pm^{(j)} = ||r_\pm^{(j)}|| = 1$, and the product state that best approximates each local Pauli expectation value can be obtained by simply normalizing $r_\pm^{(j)}$ to $\hat{r}_\pm^{(j)} = r_\pm^{(j)}/r_\pm^{(j)}$. Hence, our product-state approximation is given by $\hat{\Phi}(0)_\pm = \otimes_j |\phi_\pm^{(j)}\rangle\langle\phi_\pm^{(j)}|$ with

$$|\phi_\pm^{(j)}\rangle\langle\phi_\pm^{(j)}| = \frac{1}{2}\mathbb{1} + \hat{r}_\pm^{(j)} \cdot S_j. \qquad (25)$$

In order to construct to corresponding MPS, we solve the eigenvalue problem of $\frac{1}{2}\mathbb{1} + \hat{r}_\pm^{(j)} \cdot S_j$ and construct a product state via the local eigenstates associated with the largest eigenvalue.

**Long-time simulation using MPS representations of eigenstates.** Let us consider an MPS defined via local tensors $A^{[j]\sigma_j}$ at site $j$ (with $\sigma_j = \uparrow, \downarrow$ in our case). The expectation value of an observable $\hat{O}$ supported at

**Table 1 | Convergence of the DMRG-X algorithm**

| L | Total number of states | $\chi = 8$ | $\chi = 16$ | $\chi = 24$ | $\chi = 32$ |
|---|---|---|---|---|---|
| 10 | 900 | 51 | 11 | 6 | 6 |
| 20 | 1900 | 391 | 43 | 22 | 16 |
| 40 | 3900 | 767 | 86 | 42 | 31 |
| 80 | 7900 | 1421 | 131 | 68 | 53 |
| 160 | 15,900 | 1999 | 198 | 97 | 71 |

For each system size, the table lists the number of initial states (seeds) that have not reached a rescaled energy variance below $10^{-12}$ at a bond dimension $\chi$.

lattice site $m$ is then given by

$$\langle\psi|\hat{O}|\psi\rangle = \frac{\text{Tr}[(\prod_{j=1}^{m-1} T_1^{[j]})T_O^{[m]}(\prod_{l=m+1}^{L} T_1^{[l]})]}{\text{Tr}[\prod_{j=1}^{L} T_1^{[j]}]}, \tag{26}$$

where the local transfer operator $T_O^{[j]}$ is defined for any observable $\hat{O}$ supported at site $j$ as

$$T_O^{[j]} = \sum_{\sigma_{j_1},\sigma_{j_2}} A^{[j]\sigma_{j_1}} O_{\sigma_{j_1}\sigma_{j_2}} \left(A^{[j]\sigma_{j_2}}\right)^*. \tag{27}$$

We now discuss how to compute a local time-dependent expectation value of a state

$$|\Psi(t)\rangle = \sum_{i=1}^{r} \alpha_i e^{-iE_i t}|E_i\rangle \tag{28}$$

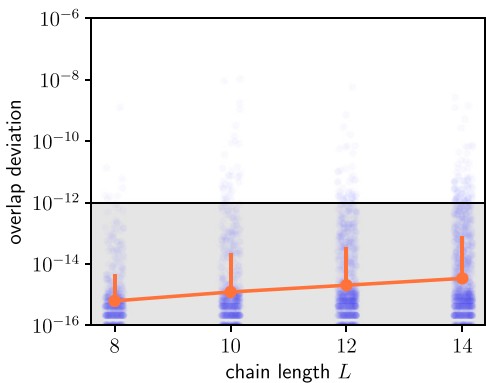

**Fig. 5 | Comparison with exact diagonalization.** For each system-size $L \in \{8, 10, 12, 14\}$, we sampled 100 disorder realizations and computed all $L$ non-trivial deformed domain-wall states per disorder realization using DMRG-X in the same way as for our main results (the algorithm terminates once $\sigma^2/E^2$ has fallen to at least $10^{-12}$). Given such a state $|\Psi_\text{MPS}\rangle$, we then obtained the closest eigenstate (in terms of overlap) via exact diagonalization $|\Psi_\text{ED}\rangle$ and computed the deviation of the overlap from unity $\delta := 1 - |\langle\Psi_\text{MPS}|\Psi_\text{ED}\rangle|$. The individual data points $\delta_i$ for the various states are shown as light dots. The orange line corresponds to the log-average: let $\mu$ and $s$ denote the mean and standard deviation of $\log(\delta_i)$. Then the orange line is given by $\exp(\mu)$ and the size of the upper error bar by $\exp(\mu+s) - \exp(\mu)$. We plot the log-average because the sample mean is strongly biased by the comparably few data points with $\delta_i \cdot 10^{-10}$, whereas the bulk of the data points lies significantly below $10^{-12}$ for every system size.

in the case where the energy eigenstates $|E_i\rangle$ are given as MPS with matrices $A_i^{[j]\sigma_j}$ and a bond dimension $\chi$. The state $|\Psi(t)\rangle$ can be expressed as an MPS with bond dimension $r\chi$ by setting

$$B^{[j]\sigma_j} = \oplus_i A_i^{[j]\sigma_j} \quad j \neq m \tag{29}$$

$$B^{[m]\sigma_m}(t) = \oplus_i \alpha_i e^{-iE_i t} A_i^{[m]\sigma_m}. \tag{30}$$

From now on, let $T_O^{[j]}$ denote the local transfer operators associated with the tensors $B^{[j]\sigma_j}$. Then the time-dependent expectation value takes the form

$$\langle\Psi(t)|\hat{O}|\Psi(t)\rangle = \frac{\text{Tr}[T_\text{left} T_O^{[m]}(t) T_\text{right}]}{\text{Tr}[T_\text{left} T_1^{[m]}(t) T_\text{right}]}, \tag{31}$$

where $T_\text{left} = \prod_{j=1}^{m-1} T^{[j]}$ and $T_\text{right} = \prod_{l=m+1}^{L} T^{[l]}$. Importantly, these left and right transfer operators are independent of $t$ and can be computed once and for all so that all time dependence is contained in the local transfer operator $T^{[m]}(t)$. Therefore, it is possible to compute local, time-dependent expectation values at arbitrary times, even for large systems. We used this technique to calculate the expectation values in Fig. 1.

## Preliminary exact-diagonalization numerics

We performed preliminary small-scale exact-diagonalization numerics targeting small sublattice entanglement, which allowed us to identify domain walls as promising seeds to construct non-equilibrating product states. This procedure consisted of the following steps for systems of sizes $L = 8, 10, 12$:

1. Sample a disorder realization.
2. Compute all energy eigenstates via exact diagonalization.
3. For each energy eigenstate $|E_j\rangle$, compute the second Rényi entropy $S_2(E_j)$ of the reduced density matrix associated with every second lattice site (sublattice entanglement).
4. Sort the energy eigenstates according to their sublattice entanglement so that $S_2(E_j) \leq S_2(E_k)$ if $j \leq k$.
5. For the $m$ eigenstates with the smallest sublattice entanglement and all pairs $(E_j, E_k)$ with $j, k = 1, \ldots, m$ and $j < k$, construct product-state approximations

$$\left|\Phi_\pm^{(j,k)}\right\rangle \approx \left|\Psi_\pm^{(j,k)}\right\rangle := \frac{1}{\sqrt{2}}\left(\left|E_j\right\rangle \pm \left|E_k\right\rangle\right) \tag{32}$$

and compute the minimum fidelity $F^{(i,j)} = \min_\pm |\langle\Phi_\pm^{(j,k)}|\Psi_\pm^{(j,k)}\rangle|$, the magnetization profile of $|\Phi_\pm^{(j,k)}\rangle$ (local expectation values of the Pauli

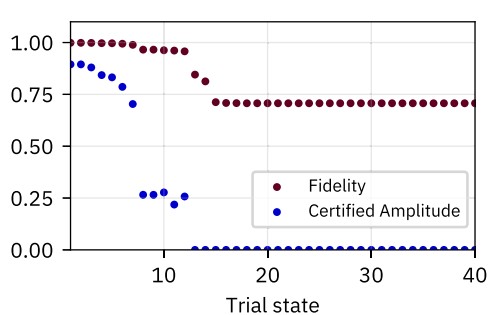

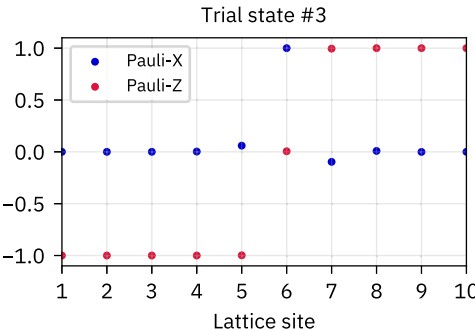

**Fig. 6 | Preliminary numerics.** Exemplary data for a single disorder realization with $W = 8$ and a system of $L = 10$ spins from our preliminary numerics. Left: Fidelities $F^{(i,j)}$ of the first 40 trial states sorted in non-increasing order and their associated certified amplitudes. Right: Magnetization profile in terms of the expectation value of

the Pauli-$X$, $Z$ operators of each lattice site for the third trial state according to the order on the left. The state has fidelity $F^{(i,j)} = 0.998$, certified amplitude $A_\text{cert.} = 0.88$ and clearly corresponds to a deformed domain wall.

operators), as well as the associated certified amplitudes. Typically, we chose $m = 20$.

6. Plot fidelities and certified amplitudes and manually inspect the magnetization profile for those states $|\Phi_{\pm}^{(j,k)}\rangle$ with large fidelities and large certified amplitudes. Exemplary data is shown in Fig. 6. The outcome of these numerics was a consistent finding of deformed domain walls with large certified amplitudes, which led to the formulation of the DMRG-X-based algorithm directly targeting deformed domain walls.

## Data availability

All our raw data, as well as the code generating the raw data and the data plots, have been deposited in the Zenodo database at https://doi.org/10.5281/zenodo.7144832 and https://doi.org/10.5281/zenodo.8245018[82,83].

## Code availability

All the code generating the raw data and the data plots from the raw data have been deposited in the Zenodo database at https://doi.org/10.5281/zenodo.7144832 and https://doi.org/10.5281/zenodo.8245018[82,83].

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

## Acknowledgements

H.W. would like to thank Merlin Füllgraf and Daniel Burgarth for useful discussions and Berislav Buča for comments on an earlier version of the manuscript. We acknowledge support by the Deutsche Forschungsgemeinschaft (DFG, German Research Foundation) through SFB 1227 (DQ-mat) (T.J.O.), Quantum Valley Lower Saxony (T.J.O.), and under Germany's Excellence Strategy EXC-2123 QuantumFrontiers 390837967 (H.W., T.J.O., C.K.). Moreover, we acknowledge support by 'Niedersächsisches Vorab' through the 'Quantum- and Nano-Metrology (QUANOMET)' initiative within the project P-1 (C.K., K.S.C.D.).

## Author contributions

H.W., T.J.O., and C.K. conceived the research problem. H.W. performed the preliminary numerical studies, worked out the analytic arguments, generated the data for Fig. 6, Supplementary Figs. 2 and 3 and drafted the manuscript. K.S.C.D. implemented the DMRG-X algorithm and generated the main data as well as the data for comparison with exact

diagonalization. H.W., T.J.O. and C.K. wrote the final manuscript. All authors regularly discussed the work and commented on the manuscript.

## Funding

## Competing interests
The authors declare no competing interests.
