## [Peer Review File · Nature Communications]

REVIEWER COMMENTS

Reviewer #1 (Remarks to the Author):

The authors construct simple initial states that exhibit many-body revivals in a localized system that is analogous to the quantum many-body scars. Their starting point is an observation that superpositions of eigenstates that approximate a product state are candidate initial states that show revivals. They numerically attempt to show that such states exist in the well known MBL model : $S=1/2$ Heisenberg chain in a random magnetic field. The knowledge of the initial state (deformed domain wall) allows them to show the existence of long time revivals using the DMRG techniques. They also argue that there exist product states with multiple dynamical excitations drawing parallels with Hilbert space-fragmented systems.

The paper is interesting and is well written conveying the results in a clear fashion. Their numerical results look sound.

I would be happy to recommend publication of this paper in Nature Communications.

I have a minor suggestion and a question that might be useful to consider discussing in the manuscript:

This work might also be interesting to experimentalists alike. I would recommend the authors to include some comments on how this work might be realizable in an experiment.

The existence of states with multiple excitations is interesting. Can the authors comment on whether these special states constitute a subspace with some emergent algebraic structure ?

Reviewer #2 (Remarks to the Author):

This is a very interesting paper with potentially significant results for the dynamics of disordered quantum systems, like spin chains, that undergo many-body localization (MBL) transition. The authors are able to show the existence of a particular type of fine-tuned product states that are well approximated by simple superpositions of a few many-body eigenstates in the MBL phase. As a result, dynamics starting from such product states show remarkable periodic revivals analogous to the so-called many-body scarring phenomena seen so far, typically for non-disordered systems with approximate emergent symmetries. This observation by the authors is contrary to the general belief that the dynamics in the MBL phase, starting from generic initial states (e.g., product states), reach a steady state (equilibrate) due to dephasing though it does not thermalize. The authors provide a clear-cut and innovative recipe to construct such product states through DMRG-X algorithm and give some analytical understanding of the result.

Overall, I find the results and the analysis in the paper very interesting and relevant for people working in disordered systems and quantum dynamics. However, I cannot wholly judge the validity of their numerical DMRG-X calculations since I do not have first-hand experience using the algorithm. I believe this has become a relatively standard numerical technique by now. In my opinion, the paper can, in principle, be suitable for Nat. Comm., but before making any judgment I would like the authors to address the following comments and questions.

1. The authors should show the comparison between the results from DMRG-X and exact diagonalization (ED) for smaller systems like system size 10-18.

(a) For example, they say that they did a structure search in ED and found the existence of pair of energy eigenstates that are well approximated by simple product states. But they do not show any results for those, neither in the main text nor in the SM. They should show these results, at least in the SM.

(b) I would like to see the comparison between DMRG-X and ED for the results of Figs.1,2 for small systems.

(c) The authors should provide the overlap between the particular eigenstates found in DMRG-X and ED for small systems.

2. On page 3, first column, first paragraph, the definition of sublattice entanglement is not clear. The statement about sublattice entanglement here is not comprehensible to me.

3. It is not clear from the argument given by the authors why deformed domain wall states do not dephase and get broadened over the localization length and simple domain wall states (ref. 73) do. It seems that the same argument should be applicable to both cases.

4. For states with multiple domain walls, as discussed in the SM, what determines that the distance between the domain walls is sufficiently large? Naively, it should be the localization length. The authors should comment.

5. Definitions of several crucial quantities, like $|\phi_{\text{pm}}^{\{1\}}\rangle$ etc., the operator A , not given in the main text. The authors should define these in the main text itself.

Reviewer #3 (Remarks to the Author):

The authors provide strong evidence for the existence of local observables that oscillate indefinitely in many-body localized (MBL) systems. As they point out, this is at variance with previous beliefs that MBL systems, though non-thermalizing, always reach local equilibria. This connects the authors' work with recent research on quantum many-body scars.

The manuscript is well written and its claims are very well supported. It presents an important result which will be of interest to a broad readership.

Minor comments:

1. I can't follow the logic in the sentence, "Since even generic translationally invariant matrix-product states (MPS) [69, 70] have extensive sub-lattice entanglement entropies [71], small sublattice entanglement indicates a closeness to product states." Closeness to a product state implies small sublattice entanglement, but why does generic MPS having large sublattice entanglement imply that a state with small sublattice entanglement must be close to a product state? How about a state which is a direct product of a highly entangled state defined on even sites and a highly entangled state defined on odd sites?

2. Around Eq. (8) I was a bit confused why $|\Psi_{\text{pm}}^{\{k\}}\rangle$ should be close to a product state. It is a sum of two DMRG-X-optimized states which were seeded with product states (which in turn sum up to a product state). But in general $|\Psi_{\text{pm}}^{\{k\}}\rangle$ could be far away from a product state. If I'm not mistaken, the closeness to a product state is simply a heuristic finding the authors made in their earlier exact diagonalization study with even/odd-bipartition rather than something that is true only for strong disorder?

3. In the Supplementary Material (SM), above Eq. (25), shouldn't the total magnetization be $-L/2+1/2$ rather than $-L+1/2$?

4. How is the standard deviation in Fig. 4 (left) of the SM plotted? It looks like it is close to 1.

5. At the bottom of page 2, SM, the rescaled energy variance should be σ^2/E^2 (rather than its inverse). Also, σ should be called standard deviation, rather than variance, in the bracket.

6. The bottom equation of page 5, SM, misses a closing bracket in the exponent $-i(\hat{H} - E)$. Also, the derivative should probably be $+i(\dots)$ rather than $-i(\dots)$.

Replies to the reviewers – “Reviving product states in the disordered Heisenberg chain”

Comments by the reviewers in black, our replies in blue.

Reviewer #1

[...]

The paper is interesting and is well written conveying the results in a clear fashion. Their numerical results look sound.

I would be happy to recommend publication of this paper in Nature Communications.

This work might also be interesting to experimentalists alike. I would recommend the authors to include some comments on how this work might be realizable in an experiment.

This is a very good point. We have added a paragraph to the Discussion section emphasizing that it should be possible to observe our predictions in present-day or near-future experiments.

The existence of states with multiple excitations is interesting. Can the authors comment on whether these special states constitute a subspace with some emergent algebraic structure ?

Due to the dependence on the precise disorder-realization we do not expect a clean algebraic structure to emerge. However, we also cannot rule it out at this time and this may be an interesting topic for future investigations. We have added a corresponding comment.

Reviewer #2

[...]

Overall, I find the results and the analysis in the paper very interesting and relevant for people working in disordered systems and quantum dynamics. However, I cannot wholly judge the validity of their numerical DMRG-X calculations since I do not have first-hand experience using the algorithm. I believe this has become a relatively standard numerical technique by now. In my opinion, the paper can, in principle, be suitable for Nat. Comm., but before making any judgment I would like the authors to address the following comments and questions.

1. The authors should show the comparison between the results from DMRG-X and exact diagonalization (ED) for smaller systems like system size 10-18.

(a) For example, they say that they did a structure search in ED and found the existence of pair of energy eigenstates that are well approximated by simple product states. But they do not show

any results for those, neither in the main text nor in the SM. They should show these results, at least in the SM.

As the outcome of the preliminary ED numerics was primarily to lead us to consider DMRG-X with domain wall seeds, but our main results are otherwise independent from these preliminary numerics, we have previously indeed left out a further description of this search. We have now added a section to the SM with a brief description of how this preliminary investigation was carried out together with exemplary data (see new Fig.8) for a system of $L=10$ spins.

(b) I would like to see the comparison between DMRG-X and ED for the results of Figs.1,2 for small systems.

(c) The authors should provide the overlap between the particular eigenstates found in DMRG-X and ED for small systems.

We have now added a figure (new Fig. 6 in the SM) comparing the particular eigenstates found using DMRG-X with ED for small systems as requested. We find that for system-sizes up to $L=14$, the deviation of the overlap from unity is always below 10^{-8} and below 10^{-12} for the bulk of the states (in fact very often below 10^{-14}). This alone already guarantees that Figs. 1,2 will be unchanged when repeated using ED. We have hence omitted re-doing these figures using ED for small systems.

We would also like to emphasize again that small standard deviations of energy (as requested by the convergence threshold for our algorithm) alone suffice to guarantee that our results are sound for time-scales at least up to the order of the inverse of the standard deviation (in our case times of the order of 10^6 in the given units).

2. On page 3, first column, first paragraph, the definition of sublattice entanglement is not clear. The statement about sublattice entanglement here is not comprehensible to me.

We acknowledge that this part of the manuscript was not formulated optimally, as also witnessed by the comment of Reviewer #3. We have now reformulated this section and provide additional information about our small-scale, preliminary, exact-diagonalization study in the SM.

3. It is not clear from the argument given by the authors why deformed domain wall states do not dephase and get broadened over the localization length and simple domain wall states (ref. 73) do. It seems that the same argument should be applicable to both cases.

We are not entirely sure whether we understood the reviewers comment correctly, but our general argument for the existence of deformed domain walls that show coherent oscillations (without dephasing) hinges on the quasi-locality of the unitary U mapping the computational basis to energy eigenstates.

This unitary cannot be exactly local (as this would lead to a non-interacting model) and therefore will necessarily prepare a deformed (and even slightly entangled) domain wall state when

applied to a simple domain wall. Therefore we do not see how our argument can be used to establish that simple domain wall states should show coherent oscillations.

4. For states with multiple domain walls, as discussed in the SM, what determines that the distance between the domain walls is sufficiently large? Naively, it should be the localization length. The authors should comment.

This can indeed be seen as a definition for the localization length. The argument we provide in terms of the quasi-local unitary implies that this distance is roughly $\sim 2 \times$ localization length. We have added a comment in the SM.

5. Definitions of several crucial quantities, like $|\phi_{\pm}^{(1)}\rangle$ etc., the operator A , not given in the main text. The authors should define these in the main text itself.

The $|\phi_{\pm}^{(1)}\rangle$ are implicitly defined by Eq. (3). We have now partly reformulated Section II.A. to emphasize the definition of the $|\phi_{\pm}^{(1)}\rangle$ and also include the technical definition of the observable \hat{A} . We believe that all relevant quantities are now fully defined in the main text.

Reviewer #3

[...]

The manuscript is well written and its claims are very well supported. It presents an important result which will be of interest to a broad readership.

Minor comments:

1. I can't follow the logic in the sentence, "Since even generic translationally invariant matrix-product states (MPS) [69, 70] have extensive sub-lattice entanglement entropies [71], small sublattice entanglement indicates a closeness to product states." Closeness to a product state implies small sublattice entanglement, but why does generic MPS having large sublattice entanglement imply that a state with small sublattice entanglement must be close to a product state? How about a state which is a direct product of a highly entangled state defined on even sites and a highly entangled state defined on odd sites?

We acknowledge that this part of the manuscript was not formulated optimally. Our aim was not to argue that small sublattice entanglement guarantees closeness to product states, but is merely a useful heuristic. The fact that even MPS, which generally have small entanglement across bipartitions, do have large sublattice entanglement suggests that targeting small sublattice entanglement from a heuristic point of view targets states with exceptionally small entanglement. We have now further provided some of the results of this preliminary search into the SM and reformulated this part of the paper to make clear that targeting small sublattice entanglement is merely a useful heuristic.

2. Around Eq. (8) I was a bit confused why $|\Psi_{\text{pm}}^{(k)}\rangle$ should be close to a product state. It is a sum of two DMRG-X-optimized states which were seeded with product states (which in turn sum up to a product state). But in general $|\Psi_{\text{pm}}^{(k)}\rangle$ could be far away from a product state. If I'm not mistaken, the closeness to a product state is simply a heuristic finding the authors made in their earlier exact diagonalization study with even/odd-bipartition rather than something that is true only for strong disorder?

Indeed, for most seeds the state $|\Psi_{\text{pm}}^{(k)}\rangle$ will be far from a product state. The closeness to a product state for certain choices of eigenstates was at first a heuristic finding from our exact diagonalization study, which led us to consider domain-wall seeds. Our main results show that for domain-wall seeds, closeness to a product state is indeed a generic case for sufficiently strong disorder. This in turn immediately implies the non-equilibrating behavior for the associated initial states. We have now clarified this in the paper.

3. In the Supplementary Material (SM), above Eq. (25), shouldn't the total magnetization be $-L/2+1/2$ rather than $-L+1/2$?

Indeed, thanks for pointing this out. Fixed now.

4. How is the standard deviation in Fig. 4 (left) of the SM plotted? It looks like it is close to 1.

It's plotted according to the scale on the left axis. For example, for $L=10$ it is of the order of 0.01 for the median fidelities and 0.005 for the maximum fidelities. We have now clarified this in the caption.

5. At the bottom of page 2, SM, the rescaled energy variance should be σ^2/E^2 (rather than its inverse). Also, σ should be called standard deviation, rather than variance, in the Bracket.

Indeed, thanks for pointing this out. Fixed now.

6. The bottom equation of page 5, SM, misses a closing bracket in the exponent $-i(\hat{H} - E)$. Also, the derivative should probably be $+i(\dots)$ rather than $-i(\dots)$.

Indeed, thanks for pointing this out. Fixed now.

REVIEWERS' COMMENTS

Reviewer #1 (Remarks to the Author):

The authors have satisfactorily answered my comments. I am happy to recommend publication of this paper in Nature Communications.

Reviewer #2 (Remarks to the Author):

The authors have satisfactorily addressed all my queries and that of the other referees. The paper reports an important discovery of a class of initial states that show periodic revival, akin to many-body scars in clean systems, in the many-body localized (MBL) phase of interacting disordered systems. The authors come up with an innovative method to construct such states and provide a nice analytical and numerical analysis of their results. Thus I recommend publication in Nature Communications.

Reviewer #3 (Remarks to the Author):

I am happy with the authors' response and modifications to the manuscript. They satisfactorily address mine and the other reviewers' comments.

Two minor things I spotted:

* The authors are inconsistent in calling their appendix "Supplemental Material" or "Appendix"

* Below Eq. (45) in the appendix, there is a "|" missing in front of $\tilde{\Psi}_{\pm}^{(k_2)}$.